# Design of a Novel and Potent Multi-Epitope Chimeric Vaccine against Human Papillomavirus (HPV): An Immunoinformatics Approach

**DOI:** 10.3390/biomedicines11051493

**Published:** 2023-05-22

**Authors:** Muhammad Shahab, Dejia Guo, Guojun Zheng, Yening Zou

**Affiliations:** 1State Key Laboratories of Chemical Resources Engineering, Beijing University of Chemical Technology, Beijing 100029, China; shahabkhan1852@gmail.com; 2Sinovac Life Sciences Co., Ltd., Beijing 102601, China; guodj8210@sinovac.com

**Keywords:** human papillomavirus, reverse vaccinology, B-Cell epitope, T-Cell epitope, molecular docking, MD simulation

## Abstract

In the current era, our experience is full of pandemic infectious agents that no longer threaten the major local source but the whole globe. One such infectious agent is HPV, a sexually transmitted disease that can cause various clinical disorders, including benign lesions and cervical cancer. Since available vaccines still need further improvements in order to enhance efficacy, our goal was to design a chimeric vaccine against HPV using an immunoinformatics approach. For designing the vaccine, the sequence of HPV was retrieved, and then phylogenetic analysis was performed. Several CTL epitopes, HTL epitopes, and LBL epitopes were all predicted using bioinformatics tools. After checking the antigenicity, allergenicity, and toxicity scores, the best epitopes were selected for vaccine construction, and then physicochemical and immunological properties were analyzed. Subsequently, vaccine 3D structure prediction, refinement, and validation were performed. Molecular docking and dynamics simulation techniques were used to explore the interactions between the Toll-like receptor 2 and the modeled vaccine construct. To ensure the vaccine protein was expressed at a higher level, the construct was computationally cloned into the pET28a (+) plasmid. The molecular docking and simulation results showed that our designed vaccine is stable, of immunogenic quality, and has considerable solubility. Through in silico immune simulation, it was predicted that the designed polypeptide vaccine construct would trigger both humoral and cellular immune responses. The developed vaccine showed significant affinity for the TLR2 receptor molecule. However, additional laboratory research is required to evaluate its safety and efficacy.

## 1. Introduction

Sexually-transmitted infections are the dominant cause of 50–70% of morbidity and mortality cases worldwide in sexually active individuals. One such infectious agent is the human papillomavirus. In reality, HPV is the most common sexually transmissible agent [1]. Sexually active males and females are susceptible to HPV infection at least once, although many of them develop no burdensome health consequences [2]. HPV is a small double-stranded DNA virus with 225 types that can be grouped into five classes, i.e., α, ß, γ, µ, and ν [3]. According to the burden of disease that it causes, HPV can be segregated into two types, low-risk HPV, which leads to the development of cutaneous and anogenital warts, and high-risk HPV, which results in oropharyngeal cancers in addition to anogenital cancers, including cervical and penile cancers [4,5] as elucidated in Appendix A. Nearly half of all malignancies that are related to infections in the world involve HPV. It was estimated that HPV was cleared by the immune system or went into dormancy within the infected patients in a year or two in 90% of cases [6]. High-risk HPV-positive women develop cervical cancer after 3–5 years after medical testing [7]. HPV is a small, non-enveloped, 60 nm-diameter, double-stranded DNA virus that belongs to the Papillomaviridae family. The DNA is a circular 7–8 kb molecule that encodes eight functional and two structural proteins in addition to the non-coding stretch, the long terminal region (LTR) [8]. The pentameric (five copies) form of L1 is anchored to a monomer of L2, and this assembly is repeated multiple times to form the 72 capsids of HPV [9]. The major capsid component, L1 (55 kDa), contains constant as well as variable regions that are responsible for surface antigenicity, the interaction with host plasma membrane receptors prior to its penetration. At the same time, these regions are the target for host antibody generation. Hence, the presence of this protein does not come as a surprise as it is responsible for the tremendously variable genotype of HPV [10]. In addition, L1 is the target for current therapeutics and vaccines owing to two reasons: Firstly, high-affinity domains in the infected host that are capable of binding it firmly and mounting the appropriate corresponding immune response. Secondly, LI has the ability to form large, non-infectious, but potent immunogenic self-assemblies [11].

Currently, three approved vaccines against HPV have demonstrated a reduction in the development and progression of cutaneous and anogenital warts and cancers. The approved vaccines are bivalent, tetravalent, and nine-valent [12]. The first one is Cervarix, manufactured by GSK and targeting HPV-16/18, which account for approximately 70% of cervical cancers. The second vaccine is Gardasil, a quadrivalent vaccine produced against HPV 6 and 11, which underlie 90% of genital warts, besides the previous targets (HPV 16 and 18). The last one is Gardasil 9, manufactured by Merck, which provides immunity towards the last two types in addition to HPV 31, 33, 45, 52, and 58, implicated in 18% of invasive cervical cancers [13,14]. Indeed, the application of these vaccines significantly reduced the incidence of cervical cancer due to HPV infection in countries like Australia and Luxembourg [1,15]. In Sweden, vaccinated women had a lower incidence of cervical cancer (47/100,000) than unvaccinated women (94/100,000) [16]. Although proven efficacious in reducing the incidence of HPV-positive cases, the currently approved vaccines still need further improvements in order to enhance their efficacy. However, reducing the negative side effects is important, as some reports document that the HPV vaccine is more immunogenic than the virus itself [17].

Immunoinformatics is one of the fastest, most precise, and most reliable approaches to developing vaccines against virulent pathogens. Given the advantages and feasibility of designing vaccines using immunoinformatics approaches, our study aimed to design a multi-epitope subunit vaccine against HPV. The present study used a wide range of antigenic proteins from HPV to design B- and T-cell epitopes. For this purpose, the MHC-II and MHC-I binding epitopes were used as a predictive method, and a vaccine composed of many epitopes was produced in the end. Additionally, molecular docking, molecular dynamics for stability profiling, and “in silico expression analysis” were employed to check the immune response reaction and stability stimulated by the final vaccine produced.

## 2. Methodology

The immunoinformatics approach for the design and analysis of the current chimeric vaccine is depicted in Figure 1.

### 2.1. Sequence Retrieval and Prioritization

The reference sequence of the human papillomavirus type-16, L2 capsid protein (Uniprot; AAA92891.1) was first retrieved from Uniprot, and then the sequence was deposited to Blastp for protein-protein Blast [1], from which 10 sequences were downloaded in FASTA format. The sequence was aligned using the Muscle v3.6 program [2]. Phylogenetic analysis of all the sequences was performed using Mega-X [3]. The potential antigenicity of precursor proteins was determined by Vaxijen ver 2.0 [4]. The protein with the highest antigenicity was considered for further analysis. Subsequently, we used the AllerTOPver2.0 web portal [5] to check the allergenicity of the proteins. The target organism was selected as a “virus” in the former tool, which used the epitope sequence as an input in the query, and a threshold score of 0.4 was specified. In addition, the secondary and tertiary structures of the protein were predicted via the PsiPred tool and I-TASSER; (https://zhanggroup.org/I-TASSER//, accessed on 5 January 2023). The structure with the highest C-score was selected to be the most authentic architecture [6].

### 2.2. Epitope Mapping

To design an epitope-based vaccine, it is essential that it can induce immunological responses. The obtained protein sequences were used as precursors for the immunogenic epitopes. We used the online IEDB conservation analysis tool to examine the conservation of expected epitopes (accessed on 5 January 2023) [7,8]. We examined the ability of the top-ranked epitopes to induce interferon (IFN) by using the IFN-epitopes tool [9]. Next, cytotoxic T-lymphocytes (CTL) and their epitopes were predicted using the NetCTL 1.2 server [10]. All of the predicted epitopes and their corresponding antigenicity, toxicity, and allergenicity were checked by VaxiJen [11,12], ToxinPred [18], and AllergenFP [12,13]. Antigens with no toxicity or allergenicity were considered for further protocols.

### 2.3. Vaccine Construct Design and Structure Prediction

The top-ranked epitopes were joined by different linkers such as GPGPG, since these linkers improve the stability of the structure by preventing self-folding, enhancing the epitope presentation [14]. Moreover, in an attempt at improving immunogenicity, the EAAAK linker was used as an adjuvant of the vaccine construct to enhance the vaccine’s efficacy. Then, the His-tag (6 his residues) was added to the construct’s C-terminal as described earlier [15]. Afterward, the vaccine’s physicochemical properties and secondary structure were predicted via the ProtParam [16] and PSIPRED web servers [17] and refined using the Galax-Refine server (https://galax.seoklab.org/refine, accessed on 5 January 2023) [19]. We used PROCHECK and ProSA-web to evaluate the quality and structure of the predicted HPV model through Ramachandran and Z-score plots before and after refinement [18,20].

### 2.4. Molecular Docking

We used the Cluspro2.0 protein-protein docking server (https://cluspro.bu.edu/, accessed on 5 January 2023) to simulate the initial interaction between the vaccine construct and the innate immune receptor TLR2. This tool is regarded to be unique in its efficiency, since it rotates the ligand 70,000 rotations to end with the position with the lowest RMSD, besides its advanced and customized options [21]. The retrieved TLR2 structure was constructed by eliminating the oligosaccharides and connected ligand molecules prior to docking, and energy was reduced. A web server called ClusPro uses billions of conformations to execute rigid-body docking on two proteins. The centers of the biggest clusters are utilized as likely models of the complex from low-energy docked structures. The Toll-like receptor 2 crystal structure was downloaded from the PDB (PDB ID; 6NIG). The protein-protein interactions were explored by the RING 3.0 tool (https://ring.biocomputingup.it/, accessed on 5 January 2023) [22].

### 2.5. Systematic Analysis of the Construct

MD simulations were run using the Amber22 force field, including different parameters during the 100 ns [23]. Subsequently, heating (up to 298 K) and equilibration (1 atm pressure) were successful. Finally, CPPTRAJ was used to extract relevant information about the behavior of the biomolecule or system, such as the RMSD and RMSF [24], which are embedded in Amber22 for the stability and flexibility of vaccine proteins. Properly folded vaccine proteins were fundamental for an effective immune response [25].

### 2.6. Codon Optimization for Expression Analysis of the Vaccine Peptide

We used the EMBOSS 6.0.1 back-transeq program to convert the vaccine construct’s primary structure into a nucleotide sequence and optimized the codons using the Java Codon Adaptation tool. This ensures the availability of most commonly used codons through the E. coli (K12 strain) expression system. Then, selections were made in order to avoid three events: (i) rho-independent termination of transcription, (ii) binding to the bacterial ribosome, and (iii) restoration of restriction endonuclease (cleavage) sites. The higher the CAI score and GC%, the more optimized the sequence. The amino acid sequence of the vaccine construct was transformed back to DNA by utilizing the backtranseq tool of the EBML-EBI web services [26]. The DNA codon sequence was optimized using the JCAT tool [27] and then inserted into the pET-28a (+) plasmid between the restriction enzyme sites Ndel and Xhol. In addition, the solubility of the overexpressed vaccine protein was assessed by Protein-Sol [28].

### 2.7. Immune Response Simulation

To check the mounted immune response against the constructed vaccine, the C-ImmSim platform (http://kraken.iac.rm.cnr.it/C-IMMSIM/, accessed on 5 January 2023) was used where a position-specific scoring matrix was employed, and all of the available options were used as default. The total number of injections was administered at 1 and 84 time hours and the random seed was kept at 12,345 for up to two weeks with volume and steps set at 10 and 100, respectively. The C-immune tool predictor was used to measure immune response magnitude [29].

## 3. Result and Analysis

### 3.1. Sequence Retrieval, Phylogenetic Analysis, and Prioritization

The study was started firstly by retrieving the protein sequence of Human papillomavirus (https://www.uniprot.org/uniprot/, accessed on 5 January 2023, AAA92891.1. fasta). Then, the protein-protein blasts (Blastp) of the top ten sequences were collected for multiple sequence alignment. Phylogenetic analysis of all these sequences was performed, as shown in Appendix A. The antigenicity and allergenicity of all of the retrieved sequences were determined by using VaxiJen v2.0 and AllerTOP v2.0. After analyzing these protein sequences, the sequence (AAV91682.1) was found to have the highest antigenicity score (0.5540) and was selected for additional study (Table 1).

### 3.2. Physicochemical Characterization

The HPV L2 capsid protein has proven its ability to significantly contribute to infection and replication inside the host cells. After estimating the viral protein’s antigenicity using the Vaxijen 2.0 web server, physiochemical properties were calculated using the ProtParam program. To enhance antigenic precision, a cutoff value of 0.5 was selected. The MEV included the addition of an adjuvant 50 s ribosomal protein from Mycobacterium tuberculosis to enhance the stability and antigenicity of the nucleoprotein, which has a molecular weight of 41,825.27 Da, comprises 365 amino acids, and has an isoelectric point (pI) of 8.52, mirroring the positively charged proteins because it is above the neutral pH of 7.0. Since the instability index (II) for this nucleoprotein was computed to be 35.85, it can be concluded that it is a stable protein. In addition, the protein is highly thermostable given its aliphatic index of 77.95.

### 3.3. Linear B-Cell Epitope

Normally, eradicating the virus from the body requires both cellular and humoral immunity. For this reason, we identified B-cell epitopes against HPV by using the IEDB web service. After the total eight epitopes were anticipated, we calculated the antigenicity and allergenicity. As a result, four epitopes were finalized for vaccine construction. Figure 2 investigates the cut-off value of 0.5, showing the epitomic and non-epitomic regions. The epitopes generated were picked based on their sequence, position, and length, as well as their favorable antigenic scores and non-allergic behavior as shown in Table 2. According to the antigenic scores, the recorded minimum value was 0.4439 and the maximum value recorded was 0.6107. The antigenicity investigation revealed the minimum value and the maximum value. However, a mean value of 0.51 was noted.

### 3.4. Prediction of Helper T Lymphocytes (HTL) Epitope

Among the 150 epitopes with an IC50 of less than 1000 as predicted with IEDB MHC I search tools, 20 epitopes were shortlisted based on their interaction with at least ten alleles. Among these 20 epitopes, nine epitopes were selected for vaccine preparation. These nine epitopes were found to be the most antigenic epitopes with non-allergenic and non-toxic properties, as depicted in Table 3.

### 3.5. Prediction of Linear B Cell Lymphocyte (LBL) Epitope

Linear B cell lymphocyte (LBL) epitopes were predicted using the IEDB Kolaskar and Tongaonkar Antigenicity method. Four LBL epitopes (non-toxic and non-allergenic), obtained from HPV on the basis of their IC50 values, are listed in Table 4. The Comb scores of the predicted epitopes as well as their corresponding antigenicity were significantly high enough to be lead candidates. The epitopes AIYYKAREMGFKHIN, APILTAFNSSHKGRI, CAIYYKAREMGFKHI, and EKWTLQDVSLEVYLT are recognized as strong binders with alleles (HLA-DRB5*01:01, HLA-DRB1*09:01, HLA-DRB1*09:01, HLA-DRB5*01:01, HLA-DRB1*07:01, and HLA-DRB1*15:01).

### 3.6. Secondary Structure Prediction and Population Coverage

The designed vaccine construct also demonstrated 56.77% helices, 10.81% sheets, and 32.42% loops as predicted by PSIPRED and I-TASSER, respectively, for their secondary and three-dimensional structures (Appendix A) [30]. The TMHMM tool was used to predict the transmembrane topology of the protein. It has been found that the first regions (residues 1–111 and 134–160) are not considered as transmembrane domains. Nevertheless, residues 1170–190 and 225–243 were discovered to lie deep within the nucleoprotein’s core, whilst the last domain (residues 303–414) was positioned in the transmembrane domain.

### 3.7. Population Coverage Analysis

The population coverage of alleles refers to the percentage of individuals in a particular population who expresses a specific set of alleles. The IEDB server predicted a global population coverage of 86% and 39% for MHCI and MHCII T-cell epitopes, respectively, with a combined coverage estimated at 92%. The highest coverage of collective epitopes is estimated for European populations (95%), followed by North America at 92%, East Asia at 81%, South Asia at 89%, Southeast Asia at 79%, Southwest Asia at 78%, East Africa at 77%, West Africa at 79%, Central Africa at 66%, North Africa at 84%, South Africa at 71%, West Indies at 88%, Central America at 21%, South America at 74%, and Oceania at 72% (Figure 3). Central America shows the lowest value whereas Eastern Europe shows a high value. Together with East Asia, Europe has the largest number of alleles in the population. The most crucial MHC-I epitopes for binding are GQVDYYGLY, KSAIVTLTY, NTTPIVHLK, and WTLQDVSLEV. Three epitopes of MHC-II alleles (AIYYKAREMGFKHIN, CAIYYKAREMGFKHI, and EKWTLQDVSLEVYLT) demonstrate considerable coverage when compared to the total world population.

### 3.8. The Proposed MEV Exhibited Admirable Qualities

The shortlisted 17 epitopes (four B-cell, nine CD8^+^, and four CD4^+^ T-cell) were joined with the aid of small peptide linkers. An immunogenic adjuvant, as well as supportive peptides, was added to assemble a 365-residue-long MEV, as shown in Figure 4. A 50S ribosomal protein is used as an adjuvant in vaccine development. To produce a specific immune response, adjuvants were combined with B-cell epitopes through different linkers. To stabilize the construct, trigger a potent immune response, and enable usage in additional purification tests, four components were added. The first step was the introduction of a 50S ribosomal protein L7/L12 adjuvant with Uniprot ID: P9WHE3. Next, linkers were employed to compartmentalize and fuse the construct’s B- and T-cell-specific epitopes. A 6x His-tag was inserted into the vaccine sequence for the purpose of protein identification and purification (at the C-terminus).

### 3.9. Vaccine 3D Structure Refinement, Validation, and Solubility Prediction

A three-dimensional (3D) structure of the protein from the vaccine sequence was obtained from the trRosetta server. The Galaxy Refine server was subsequently utilized for the refinement of the protein’s 3D structure. The Galaxy Refine server made the protein structure more stable and increased its quality score in the SAVES server. Ramachandran plot analysis has revealed that 91.4% of residues in the 3D structure were in the most favored region for the refined structure. Similarly, the overall quality factor in the ERRAT program was increased to 84.53. The 3D model of the refined vaccine structure was visualized and rendered with PyMol 2 (Figure 5).

The protein-sol was used to calculate the solubility of our HPV vaccine (https://protein-sol.manchester.ac.uk/results/solubility/run-a530cad3673133be9ea5/results.html, accessed on 5 January 2023). Solubility of the inclusion body proteins is an extraordinary step in vaccine biotechnology since soluble proteins facilitate their downstream isolation and purification. The vaccine protein construct is found to be soluble (calculated score 0.4) when overexpressed in *E. coli*, as shown in Appendix A. This ensures the ease of downstream isolation and purification steps for the overexpressed vector.

### 3.10. Molecular Docking Studies between the Vaccine Construct and the TLR2 Receptor

To determine the interaction between the designed vaccine and the ligand binding pocket of the TLR2, molecular docking was carried out. In total, ten models were obtained using Cluspro for docking analysis. Among the ten models, the first model showed a good H-bond interaction as shown in Figure 6. The results from the H-bond analysis indicated that specific residue pairs interacted with each other, including HIS153-ASP453, HIS246-ARG234, ARG87-TYR244, ARG150-ASP453, LYS146-ASP502, LYS146-GLN430, ASN171-TRP550, and HIS266-LEU293, with distances of 1.73, 2.32, 2.25, 2.29, 2.11, 3.59, 1.98, and 3.19, respectively. The lowest free energy of binding between the two proteins was −1369.6 kcal/mol. PDBePISA and PDBsum have used a representation of different interactions. This indicates the highly immunogenic potency of the constructed vaccine, as demonstrated by its strong binding to the innate immune cell receptor TLR2.

### 3.11. Molecular Dynamics Simulation

The MD simulations performed for the constructed vaccine revealed its stable nature as reflected by the RMSD diagram since this stability is accomplished over 100 ns. The simulation’s output trajectory’s first frame coordinates were considered a reference while calculating the Cα-based RMSD. The protein-protein complex’s RMSD was stabilized with an average of 10 Å throughout the simulation. This emphasized the folding stability as well as the almost sustained behavior of the vaccine construct. Furthermore, the residual fluctuations were recorded, where the output trajectory’s first frame was considered a reference structure to calculate the RMSF. The residues positioned between 75–100 fluctuate the most with an average RMSF value of 10 Å. These residues are involved in direct contact with the TLR2 receptor and have a confirmational shift. Furthermore, the interacting residues of the vaccine construct with the TLR2 receptor, including 135–140, 200–220, 190–320, and 420–435, show fluctuation (RMSF average 5 Å) during the 100 ns simulation time. In addition, marked RMSF was seen at the end, which may account for the highly flexible loops at both terminals (N-terminal and C-terminal) (Figure 7A,B).

### 3.12. C-IMM Simulation

Using the C-IMMSIM immune tool, we analyzed immune responses for the assessment of the immunogenicity profile of the vaccine. It is noted that the injected antigen, after getting higher antigen counts at day five, was then neutralized slowly until day fifteen for the constructed vaccine. After antigen administration, high levels of various antibodies (IgG + IgM > 700,000; IgM > 600,000; IgG1 + IgG2; IgG1 > 500,000) were detected, and the antigen concentration decreased. Figure 8A shows that the ratios of IgG and IgM titers were substantially higher. In addition, after each immunization, the level of B-cells increased (Figure 8B). Interestingly, the effect of T-cells dramatically increased after primary and secondary immunization and was boosted at later stages (Figure 8C,D). Notably, we observed the effect of the vaccine construct on innate immune cell populations (Figure 8E). Furthermore, it should be observed that IFN-γ, IL-2, and TH cells significantly increased, which is crucial for the immune response (Figure 8F).

### 3.13. Vector Preparation and Cloning

Codon content was diminished upon codon optimization using the JCat tool from 57% to 50%. The CAI value of the optimized sequence was 0.39, indicating an acceptable expression probability in the *E. coli* K12 expression system. The optimized sequence (6373 bases) was inserted into the region between the restriction enzymes Ndel and Xhol of the plasmid, as illustrated in Appendix A. This provides a ready-to-use plasmid, pET-28a (+), containing the vaccine construct to be applied in a wet-lab setting.

## 4. Discussion

Vaccination is currently the most effective against viral treatment [31]. Conventional vaccine development has a higher failure rate, which is coupled with time and financial burdens [32]. With the advent and huge developments in bioinformatics and immunoinformatics approaches, substantial time, effort, and quality were obtained for creating peptide-based vaccinations [33]. Numerous reports utilized such an approach to design novel and potent vaccines against a wide array of bacterial, viral, and parasitic strains. They not only targeted B-cells for generating counteracting antibodies, but also T-cells to create memory cells that can memorize the designed vaccine through interacting firmly with major histocompatibility complexes (MHC) with different alleles [30,34,35,36]. Our current research aimed to create a highly effective MEV targeting HPV. As B-cells produce antibodies and assist T-cells in establishing durable adaptive immunity against viral infections, the capacity for vaccine design to develop long-lasting protection is critical. The epitope sequences were identified as potential antigens, indicating their immunogenic potential. However, vaccine development faces a major challenge in inducing allergenicity, which causes allergic reactions rather than immune responses. Given the fact that vaccines demonstrated powerful potency against widespread viral outbreaks, multiple HPV vaccines were manufactured, tested, and proved efficacious for reducing HPV infections and progression. Therefore, the search for a more effective and safer HPV vaccine covering related cancers is urgently needed. Given that there is no available vaccine for such a pandemic, this is aggravating the situation since the medical society has not yet recovered [37]. Immunoinformatics has proven itself to be a significant and powerful tool that many researchers use to test the antigenicity of certain B-cell and T-cell epitopes. In addition, complete in silico construction, testing efficacy, and cloning save both money and time, particularly in critical periods such as the current pandemic of HPV. Immunoinformatics approaches employed to date deal with only one pathogen at a time but no chimeric vaccine has been constructed against HPV. Hence, this was the goal of the present study. The potential epitopes were selected to be used further for vaccine construction. The Helper T Lymphocyte (HTL) epitopes selected proved their activity as IFN-inducers, and all of the selected epitopes were probable antigens as described by the VaxiJen 2.0 server. As the epitopes are non-toxic and non-allergenic, this confirms their safety. After joining using specific peptide linkers, the secondary and tertiary structures of the proteins were predicted and validated. The stability of the vaccine construct and the TLR-2-docked complex was validated using MD simulation. The results showed stable molecular interactions between the vaccine and the immunological receptor, ensuring the molecular stability of the multi-epitope vaccine complex in a cellular environment. Afterward, the vaccine sequence was cloned into the highly overexpressed plasmid pET-281 (+), and its solubility upon isolation and purification was acceptable [38]. Lastly, the immune simulation profile gave excellent findings involving high numbers of B-cells, Helper T Lymphocytes (HTL), CTL, NK cells, and macrophages, in addition to the predominant IFN-secretion, which provides antiviral activity. It has been documented that IFN- and NK cells are the first-line defenses against invading viruses including HPV, reflecting their important role in virus eradication [39]. All these findings emphasize the potency, efficacy, safety, and thus the eligibility of the chimeric vaccine construct as an untested strategy to limit the simultaneous pandemics. Many previous works have reported the application of immunoinformatics tools to design and test the potency of the vaccine against many targets [15,33,38]. Future clinical studies are recommended to assess the efficacy of this vaccine further.

## 5. Concluding Remarks

The aim of our work was to create a novel multi-epitope-based vaccine against HPV in order to create the most suitable vaccine that is symptom-free and risk-free. Several B- and T-cell epitopes were analyzed and combined with proper adjuvants and linkers to improve the immunization of the new vaccine. According to the results, the vaccine had good antigenicity, solubility, allergenicity, tertiary structure, and physicochemical properties. This study simulated the molecular dynamics and binding of the vaccine and TLR-2. Furthermore, the cellular immune response to antigens was confirmed via immunoassay. The results obtained in this study were experimentally valuable, therefore they can be used in developing an experimental vaccine against HPV.

## Figures and Tables

**Figure 1 biomedicines-11-01493-f001:**
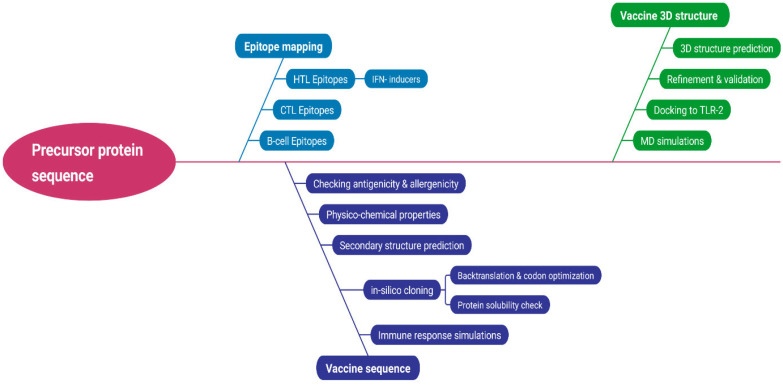
Workflow of the immunoinformatics approach used in this study.

**Figure 2 biomedicines-11-01493-f002:**
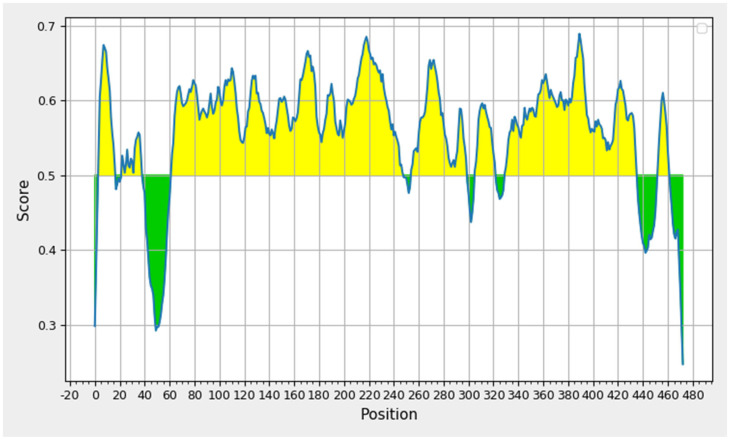
Illustration of the non-epitomic section (green) and the epitomic region of the B-cell (yellow).

**Figure 3 biomedicines-11-01493-f003:**
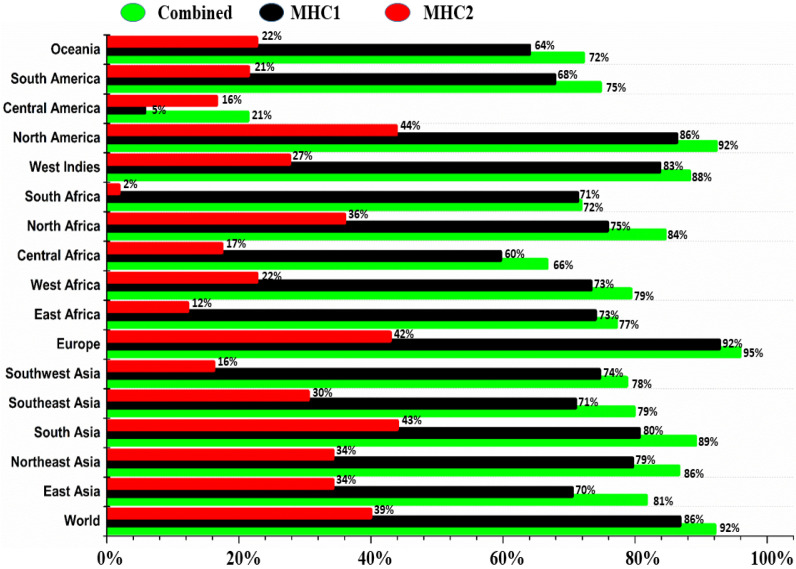
Population coverage was estimated for the selected epitopes based on MHC-I and MHC-II restriction data from all continents.

**Figure 4 biomedicines-11-01493-f004:**
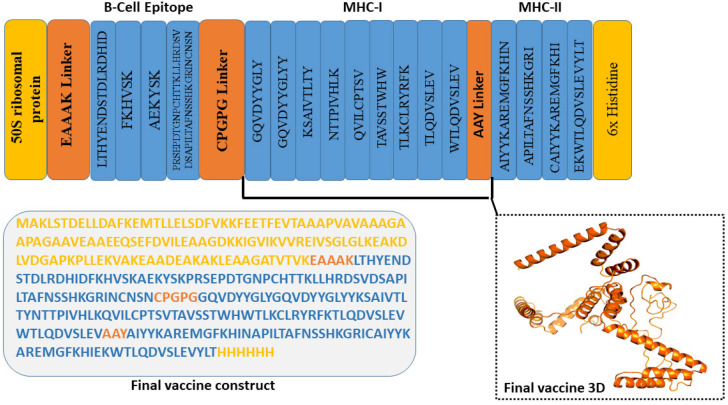
A visual representation of the multi-epitope vaccine construct (MEV) is depicted through graphics. The components of the construct are linked together using different linkers, namely (1) EAAAK, (2) CPGPG, and (3) AAY. CD8+ epitopes are joined using the AAY linker, CD4+ epitopes are joined using the CPGPG linker, and B-cell epitopes are joined using the EAAAK linker.

**Figure 5 biomedicines-11-01493-f005:**
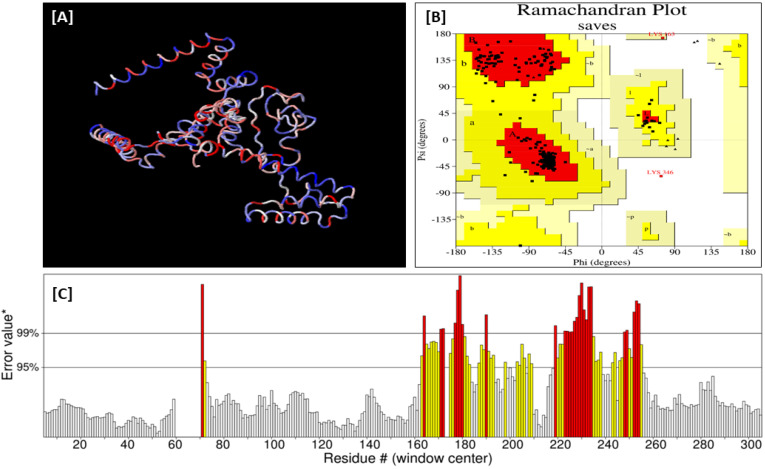
(**A**,**B**) the validated 3D model (refined) of the predicted vaccine validated by a Ramachandran plot of the PROCHECK program and ERRAT (**C**) of the SAVES server.

**Figure 6 biomedicines-11-01493-f006:**
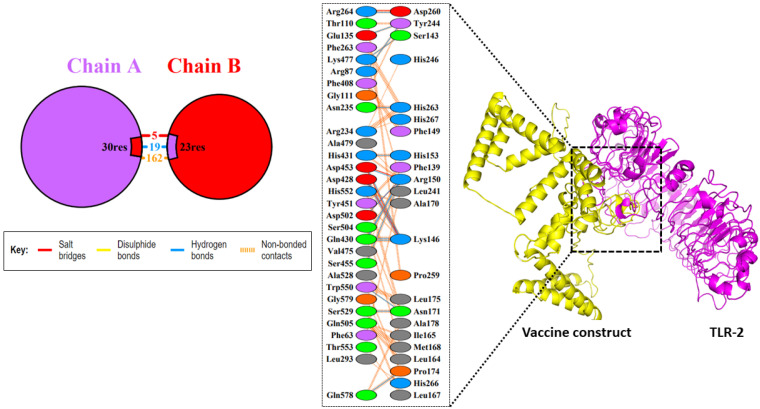
The docked complex of the designed vaccine (yellow) and TLR2 (magenta).

**Figure 7 biomedicines-11-01493-f007:**
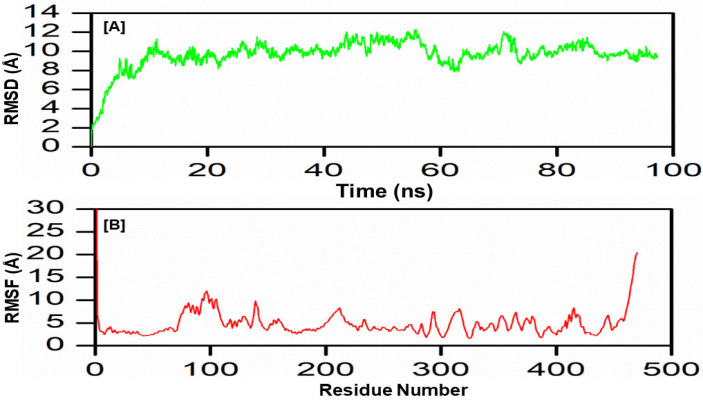
MD simulation findings of the vaccine construct for 100 ns. RMSD (**A**) and RMSF (**B**) were elucidated.

**Figure 8 biomedicines-11-01493-f008:**
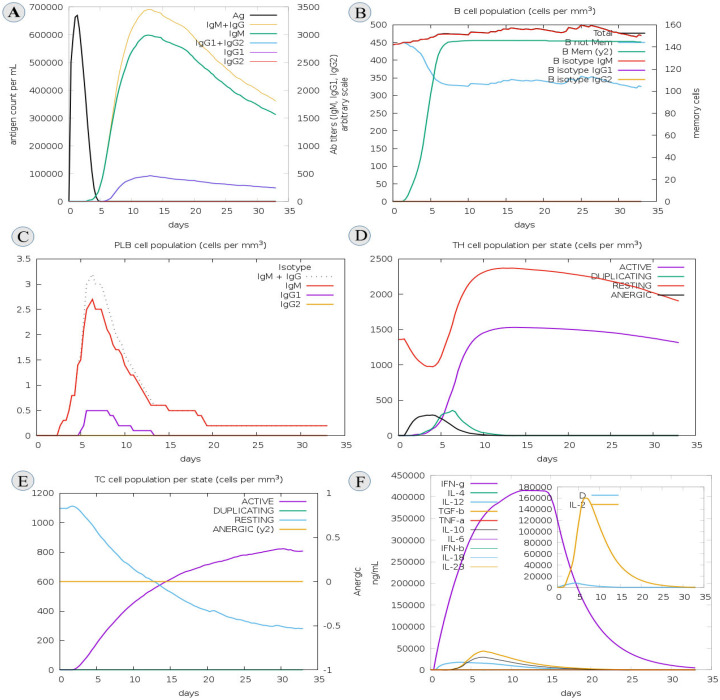
Depicts an immune simulation of vaccine constructs (**A**) Immunoglobulin production shown by black lines after antigen injection; colored lines indicate immune cell classes (**B**) Changes in B-cell population and memory production (**C**) Total number of B-lymphocyte cells in plasma per cell (**D**) Production of helper T-cells (**E**) Total number of TC cells (**F**) Displayed elevated rates of cytokines and interleukins.

**Table 1 biomedicines-11-01493-t001:** Representation of retrieved sequences along with their score.

Accession No.	Protein Name	Sequence	VaxiJen Score	Antigenicity
AAA92891.1	HPV-16 L2 capsid protein	MRHKRSAKRTKRASATQLYKTCKQAGTCPPDIIPKVEGKTIADQILQYGSMGVFFGGLGIGTGSGTGGRTGYIPLGTRPPTATDTLAPVRPPLTVDPVGPSDPSIVSLVEETSFIDAGAPTPVPSIPPDVSGFSITTSTDTTPAILDINNTVTTVTTHNNPTFTDPSVLQPPTPAETGGHFTLSSSTISTHNYEEIPMDTFIVSTNPNTVTSSTPIPGSRPVARLGLYSRTTQQVKVVDPAFVTTPTKLITYDNPAYEGIDVDNTLYFPSNDNSINIAPDPDFLDIVALHRPALTSRRTGIRYSRIGNKQTLRTRSGKSIGAKVHYYYDLSTINPAEEIELQTITPSTYTTPSHAASPTSINNGLYDIYADDFITDTFTTPVPSIPSTSLSGYIPANTTIPFGGAYNIPLVSGPDIPINTTDQTPSLIPIVPGSPQYTIIADGGDFYLHPSYYMLRKRRKRLPYFFSDVSLAA	0.6019	ANTIGEN
AAO15711.1	HPV-16 putative minor capsid protein L2	MRHKRSAKRTKRASATQLYKTCKQAGTCPPDIIPKVEGKTIADQILQYGSMGVFFGGLGIGTGSGTGGRTGYIPLGTRPPTATDTLAPVRPPLTVDPVGPSDPSIVSLVEETSFIDAGAPTPVPSIPPDVSGFSITTSTDTTPAILDINNTVTTVTTHNNPTFTDPSVLQPPTPAETGGHFTLSSSTISTHNYEEIPMDTFIVSTNPNTVTSSTPIPGSRPVARLGLYSRTTQQVKVVDPAFVTTPTKLITYDNPAYEGIDVDNTLYFASNDNSINIAPDPDFLDIVALHRPALTSRRTGIRYSRIGNKQTLRTRSGKSIGAKVHYYYDLSTINPAEEIELQTITPSTYTTPSHAASPTSINNGLYDIYADDFITDTFTTPVPSIPSTSLSGYIPANTTIPFGGAYNIPLVSGPDIPINTTDQTPSLIPIVPGSPQYTIIADGGDFYLHPSYYMLRKRRKRLPYFFSDVSLAA	0.6078	ANTIGEN
AAO85414.1	HPV-16	MRHKRSAKRTKRASATQLYKTCKQAGTCPPDIIPKVEGKTIADQILQYGSMGVFFGGLGIGTGSGTGGRTGYIPLGTRPPTATDTLAPVRPPLTVDPVGPSDPSIVSLVEETSFIDAGAPTPVPSIPPDVSGFSITTSTDTTPAILDINNTVTTVTTHNNPTFTDPSVLQPPTPAETGGHFTLSSSTISTHNYEEIPMDTFIVSTNPNTVTSSTPIPGSRPVARLGLYSRTTQQVKVVDPAFVTAPTKLITYDNPAYEGIDVDNTFYFPSNDNSINIAPDPDFLDIVALHRPALTSRRTGIRYSRIGNKQTLRTRSGKSIGAKVHYYYDLSTINPAEEIELQTITPSTYTTTSHAASPTSINNGLYDIYADDFITDTVTTPVPAIPSTSLSGYIPANTTIPFGGAYNIPLVSGPDIPINTTDQTPSLIPIVPGSPQYTIIADGGDFYLHPSYYMLRKRRKRLPYFFSDVSLAA	0.6153	ANTIGEN
AAQ10718.1	HPV-16	MRHKRSAKRTKRASATQLYKTCKQAGTCPPDIIPKVEGKTIADQILQYGSMGVFFGGLGIGTGSGTGGRTGYIPLGTRPPTATDTLAPVRPPLTVDPVGPSDPSIVSLVEETSFIDAGAPTSVPSIPPDVSGFSITTSTDTTPAILDINNTVTTVTTHNNPTFTDPSVLQPPTPAETGGHFTLSSSTISTHNYEEIPMDTFIVSTNPNTVTSSTPIPGSRPVARLGLYSRTTQQVKVVDPAFVTTPTKLITYDNPAYEGIDVDNTLYFPSNDNSINIAPDPDFLDIVALHRPALTSRRTGIRYSRIGNKQTLRTRSGKSIGAKVHYYYDLSTIDPAEEIELQTITPSTYTTTLHAASPTSINNGLYDIYADDFITDTSTTPVPSVPSTSLSGYIPANTTIPFGGAYNIPLVSGPDIPINITDQAPSLIPIVPGSPQYTIIADAGDFYLHPSYYMLRKRRKRLPYFFSDVSLAA	0.6352	ANTIGEN
AAQ10726.1	HPV-16	MRHKRSAKRTKRASATQLYKTCKQAGTCPPDIIPKVEGKTIADQILQYGSMGVFFGGLGIGTGSGTGGRTGYIPLGTRPPTATDTLAPVRPPLTVDPVGPSDPSIVSLVEETSFIDAGAPTPVPSIPPDVSGFSITTSTDTTPAILDINNTVTTVTTHNNPTFTDPSVLQPPTPAETGGHFTLSSSTISTHNYEEIPMDTFIVSTNPNTVTSSTPIPGSRPVARLGLYSRTTQQVKVVDPAFVTTPTKLITYDNPAYEGIDVDNTLYFPSNDNSINIAPDPDFLDIVALHRPALTSRRTGIRYSRIGNKQTLRTRSGKSIGAKVHYYYDLSTINPAEEIELQTITPSTYTTASHAASPTSINNGLYDIYADDFITDTSTTPVPSIPSTSLSGYIPANTTIPFGGAYNIPLVSGPDIPINTTDQTPSLIPIVPGSPQYTIIADGGDFYLHPSYYMLRKRRKRLPYFFSDVSLAA	0.6197	ANTIGEN
AAV91650.1	HPV-16	MRHKRSAKRTKRASATQLYKTCKQAGTCPPDIIPKVEGKTIADQILQYGSMGVFFGGLGIGTGSGTGGRTGYIPLGTRPPTATDTLAPVRPPLTVDPVGPSDPSIVSLVEETSFIDAGAPTPVPSIPPDVSGFSITTSTDTTPAILDINNTVTTVTTHNNPTFTDPSVLQPPTPAETGGHFTLSSSTISTHNYEEIPMDTFIVSTNPNTVTSSTPIPGSRPVARLGLYSRTTQQVKVVDPAFVTAPTKLITYDNPAYEGIDVDNTFYFPSNDNSINIAPDPDFLDIVALHRPALTSRRTGIRYSRIGNKQTLRTRSGKSIGAKVHYYYDLSTINPAEEIELQTITPSTYTPTSHAASPTSINNGLYDIYADDFITDTVTTPVPAIPSTSLSGYIPANTTIPFGGAYNIPLVSGPDIPINTTDQTPSLIPIVPGSPQYTIIADGGDFYLHPSYYMLRKRRKRLPYFFSDVSLAA	0.6090	ANTIGEN
ALB35319.1	HPV-16	MRHKRSAKRTKRASATQLYKTCKQAGTCPPDIIPKVEGKTIADQILQYGSMGVFFGGLGIGTGSGTGGRTGYIPLGTRPPTATDTLAPVRPPLTVDPVGPSDPSIVSLVEETSFIDAGAPTSVPSIPPDVSGFSITTSTDTTPAILDINNTVTTVTTHNNPTFTDPSVLQPPTPAETGGHFTLSSSTISTHNYEEIPMDTFIVSTNPNTVTSSTPIPGSRPVARLGLYSRTTQQVKVVDPAFVTTPTKLITYDNPAYEGIDVDNTLYFSSNDNSINIAPDPDFLDIVALHRPALTSRRTGIRYSRIGNKQTLRTRSGKSIGAKVHYYYDFSTIDPAEEIELQTITPSTYTTTSHAASPTSINNGLYDIYADDFITDTSTTPVPSVPSTSLSGYIPANTTIPFGGAYNIPLVSGPDIPINITDQAPSLIPIVPGSPQYTIIADAGDFYLHPSYYMLRKRRKRLPYFFSDVSLAA	0.6401	ANTIGEN
AAV91690.1	HPV-16	MRHKRSAKRTKRASATQLYKTCKQAGTCPPDIIPKVEGKTIADQILQYGSMGVFFGGLGIGTGSGTGGRTGYIPLGTRPPTATDTLAPVRPPLTVDPVGPSDPSIVSLVEETSFIDAGAPTSVPSIPPDVSGFSITTSTDTTPAILDINNTVTTVTTHNNPTFTDPSVLQPPTPAETGGHFTLSSSTISTHNYEEIPMDTFIVSTNPNTVTSSTPIPGSRPVARLGLYSRTTQQVKVVDPAFITTPTKLITYDNPAYEGIDVDNTLYFSSNDNSINIAPDPDFLDIVALHRPALTSRRTGIRYSRIGNKQTLRTRSGKSIGAKVHYYYDFSTIDPAEEIELQTITPSTYTTTSHAASPTSINNGLYDIYADDFITDTSTTPVPSVPSTSLSGYIPANTTIPFGGAYNIPLVSGPDIPINITDQAPSLIPIVPGSPQYTIIADAGDFYLHPSYYMLRKRRKRLPYFFSDVSLAA	0.6393	ANTIGEN
AAV91674.1	HPV-16	MRHKRSAKRTKRASATQLYKTCKQAGTCPPDIIPKVEGKTIADQILQYGSMGVFFGGLGIGTGSGTGGRTGYIPLGTRPPTATDTLAPVRPPLTVDPVGPSDPSIVSLVEETSFIDAGAPTPVPSIPPDVSGFSITTSTDTTPAILDINNTVTTVTTHNNPTFTDPSVLQPPTPAETGGHFTLSSSTISTHNYEEIPMDTFIVSTNPNTVTSSTPIPGSRPVARLGLYSRTTQQVKVVDPAFVTAPTKLITYDNPAYEGIDVDNTFYFPSNDNSINIAPDPDFLDIVALHRPALTSRRTGIRYSRIGNKQTLRTRSGKSIGAKVHYYYDLSTINPAEEIELQTITPSTYTPTSHAASPTSINNGLYDIYADDFITDTVTTPVPAIPSTSLSGYIPANTTIPFGGAYNIPLVSGPDIPINTTDQTPSLIPIVPGSPQYTIIADGGDFYLHPSYYMLRKRRKRLPYFFSDVSLAA	0.6090	ANTIGEN
AAV91682.1	HPV-16	MRHKRSAKRTKRASATQLYKTCKQAGTCPPDIIPKVEGKTIADQILQYGSMGVFFGGLGIGTGSGTGGRTGYIPLGTRPPTATDTLAPVRPPLTVDPVGPSDPSIVSLVEETSFIDAGAPTSVPSIPPDVSGFSITTSTDTTPAILDINNTVTTVTTHNNPTFTDPSVLQPPTPAETGGHFTLSSSTISTHNYEEIPMDTFIVSTNPNTVTSSTPIPGSRPVARLGLYSRTTQQVKVVDPAFVTTPTKLITYDNPAYEGIDVDNTLYFSSNDNSINIAPDPDFLDIVALHRPALTSRRTGIRYSRIGNKQTLRTRSGKSIGAKVHYYYDFSTIDPAEEIELQTITPSTYTTTSHAASPTSINNGLYDIYADDFITDTSTTPVPSVPSTSLSGYIPANTTIPFGGAYNIPLVSGPDIPINITDQAPSLIPIVPGSPQYTIIADAGDFYLHPSYYMLRKRRKRLPYFFSDVSLAA	0.6493	ANTIGEN

**Table 2 biomedicines-11-01493-t002:** The linear epitopes extracted along with their peptide, length, start, and end point.

Start	End	Peptide	Length
4	17	KRSAKRTKRASATQ	16
63	68	FKHVSK	6
175	180	AEKYSK	6
196	239	PRSEPDTGNPCHTTKLLHRDSVDSAPILTAFNSSHKGRINCNSN	44

**Table 3 biomedicines-11-01493-t003:** Predicted three HTL with their antigenic score and allergenic and toxic properties.

Epitopes	Interacting Alleles	Antigenicity	Allergenicity	Toxicity
GQVDYYGLY	HLA-B*15:01,HLA-A*30:02	0.6511	NO	NO
GQVDYYGLYY	HLA-B*15:01,HLA-A*01:01	0.549	NO	NO
KSAIVTLTY	HLA-B*58:01,HLA-A*30:02, HLA-A*32:01,HLA-B*15:01	0.8082	NO	NO
NTTPIVHLK	HLA-A*68:01,HLA-A*11:01	1.5790	NO	NO
QVILCPTSV	HLA-A*68:02,HLA-A*02:03	0.528	NO	NO
TAVSSTWHW	HLA-B*58:01, HLA-B*53:01,HLA-B*57:01	1.1083	NO	NO
TLKCLRYRFK	HLA-A*31:01, HLA-A*03:01,HLA-A*11:01	1.2871	NO	NO
TLQDVSLEV	HLA-A*02:03, HLA-A*02:01,HLA-A*02:06	1.3187	NO	NO
WTLQDVSLEV	HLA-A*02:06, HLA-A*02:01,HLA-A*02:03,HLA-A*68:02	1.6596	NO	NO

**Table 4 biomedicines-11-01493-t004:** LBL epitopes extracted from both viruses along with their antigenicity, toxicity, and allergenicity.

Epitopes	Interacting Alleles	Antigenicity	Allergenicity	Toxicity
AIYYKAREMGFKHIN	HLA-DRB5*01:01,HLA-DRB1*09:01	1.4971	NO	NO
APILTAFNSSHKGRI	HLA-DRB1*07:01, HLA-DRB1*15:01,HLA-DRB5*01:01	0.5116	NO	NO
CAIYYKAREMGFKHI	HLA-DRB1*04:01, HLA-DRB1*07:01	1.3288	NO	NO
EKWTLQDVSLEVYLT	HLA-DPA1*02:01/ HLA-DRB1*01:01,HLA-DPA1*03:01/DPB1*04:02	0.9309	NO	NO

## Data Availability

All data generated or analyzed during this study are included in the article.

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
