# Peer review of "Design of a Novel and Potent Multi-Epitope Chimeric Vaccine against Human Papillomavirus (HPV): An Immunoinformatics Approach"

_biomedicines, 2023, doi:10.3390/biomedicines11051493_

Round 1
Reviewer 1 Report
The manuscript describes an in silico procedure for the design of a multi-epitope vaccine against human papillomavirus (HVP). The approach includes several steps, applying a number of different in silico tools.
It is not clear why some steps are applied. There is missing or limited information about some methods, e.g. molecular docking procedure. The manuscript needs to be précised before being accepted for publication.
Here are my remarks on the manuscript:
1. In the Abstract section on Row 13 authors stated:” Since neither approved antiviral drugs nor licensed active vaccines are yet available.” However, in the Introduction section on row 60, the authors actually give the opposite information: “Currently, three approved vaccines against HPV have demonstrated a reduction in developing and progressing cutaneous and anogenital warts and cancers. The approved vaccines are bivalent, tetravalent, and 9-valent…”.
2. There are some steps in the procedure described in the manuscript, which are unclear:
Why among all proteins from HVP, the L2 capsid protein (Uniprot; AAA92891) is chosen?
Why protein-protein blast (Blastp) of the top 10 sequences was collected for multiple sequence alignment? What information was collected from this step and how it is used?
What information was achieved after phylogenetic analysis of all the sequences and how it is used for the aim of the study?
3. The reference for the AllerTOPver2.0 server is not correct.
4. In the section Results and Analysis, subsection Physicochemical characterization, the authors mention HPV nucleoprotein as significantly contributing to infection and replication inside the host cells. There is no information given in the Methodology section about HPV nucleoprotein. The only information is about L2 capsid protein. These are different proteins with different functions in the virus life cycle.
5. The description of the reverse vaccinology as a subsection need to be removed in the Introduction section: The authors did not start their procedure from the nucleotide sequence of the HPV.
6. The title of Table 2: “The linear epitope extracted along with their antigenicity, toxicity and allergenicity”, does not correspond to its content. There is no data about the antigenicity, toxicity, and allergenicity of the peptides in the table, only their starting and ending positions, sequence, and length.
7. There are abbreviations without explanations: HTL and CTL epitopes.
8. On row 262 authors claim: “When adjuvant and B-cell epitope are connected by EAAAk linker, a specific immunological response is elicited.” How authors reach to this conclusion?
9. The data from molecular docking are not shown. The authors comment on an analysis of 10 models, but there is no data about these models.
10. The authors conclude that “…docking TLR2 revealed the high probability of mounting an immune response upon in vitro testing since the docking score was -1369.6 kcal/mol”. How the exact value of the docking scores can be a criterion for stable binding? What is the threshold in this case?
11. The authors' discussion of the results from the MD simulations is not detailed enough. The fluctuations of about 4 angstroms in RMSD do not correspond to very high stability. The same conclusion can be made for the fluctuations in RMSF.
In summary, the authors of the manuscript have applied plenty of different methods used in modern drug design, but the results need to be presented and commented on in detail.
The manuscript needs revision for English language, especially the grammar.
Author Response
Dear Reviewer! please find the attached response letter.
thank you for your time and efforts.

Reviewer 2 Report
In the manuscript (ID: biomedicines-2376147) the authors to utilize an immunoinformatics approach to design a chimeric vaccine against HPV. The research studies are interesting however there is a lack of experimental studies. Therefore such a vaccine cannot be expected to be effective, although bioinformatics analysis may predict it. Further studies are needed.
In the manuscript the authors should take under consideration the following points:
1) The authors included a description of HPV and approved vaccines against HPV in the introduction. However, they didn’t emphasize the aim of the study.
2) The conclusions should indicate what is the result of the work and here it is missing.
3) How was predicted the binding site of one protein to another? What aminoacids are most important for interaction between vaccine and protein?
Author Response
Dear reviewer! please find the attached response letter.

Round 2
Reviewer 1 Report
My remarks about MD simulations are not considered.
There are two graphs in Figure 7: The plot RMSD vs time (7A) and the plot RMSF vs residue number (7B).
1. It is not known how RMSD was calculated. Usually, RMSD is calculated according to a referent structure from crystallographic data. It is not clear what is the reference structure for RMSD calculations.
2. The authors observed “little fluctuation” between 75 and 100 ns. These “little fluctuations” are higher than 4 Angstroms, which do not correlate with the conclusion:
“The results showed stable molecular interactions between the vaccination and immunological receptor, ensuring the molecular stability of the multi-epitope vaccine complex in a cellular environment.”
Such fluctuations require a longer time for MD simulations to confirm the stability of the complex.
3. The authors did not comment on the second plot RMSF vs residue number (7B) in detail. Here again, the question about the reference structure is of great importance.
There are more than 10 Angstroms fluctuations for residues between 90 and 100; high fluctuations for residues around 140 and 310 and 410. These fluctuations are not commented on.
Minor editing of the English language is required. There are meaningless sentences e.q. “The H-bond result revealed the interaction formed between the residues are.” (row 292).
Author Response
Dear reviewer! kindly check the attached file.
